# Impact of the COVID-19 Pandemic on Dermatology Care in the Chilean Public Health Sector

**DOI:** 10.3390/healthcare11050633

**Published:** 2023-02-21

**Authors:** Diego Aragón-Caqueo, Gabriel Aedo, Javier Suárez, Claudio Toloza, Antonio Guglielmetti

**Affiliations:** 1Escuela de Medicina, Universidad de Tarapacá, Arica 1000000, Chile; 2Facultad de Ciencias Médicas, Universidad de Santiago de Chile, Santiago 8320000, Chile; 3Escuela de Medicina, Universidad de Valaparaíso, Viña del Mar 2520000, Chile

**Keywords:** COVID-19, pandemic, dermatological consultations, public health

## Abstract

Due to the Coronavirus-19 (COVID-19) pandemic, most resources of the public health system were allocated to the increasing demand from respiratory patients. From this, it is expected that specialty consultations would decrease drastically. Access to dermatology care in the Chilean public health has been historically limited. To evaluate the impact of the pandemic on dermatology care, the total number of dermatological consultations (DCs) to the Chilean public sector in 2020 is analyzed according to sex and age range and compared with the available databases from 2017 to 2019. From this, 120,095 DCs were performed during 2020, with an incidence of 6.3 consultations per 1000 inhabitants. When compared to 2019 (*n* = 250,649), there was a 52.1% decrease. The regions most affected were located in the central part of Chile, which correlates with the regions most affected by the pandemic. Age and sex distributions remained similar to previous years but lower in amplitude. The month with the lowest number of consultations was April, with a gradual increase observed thereafter until December 2020. Although DCs decreased drastically in the Chilean public sector during 2020, sex and age range proportions were conserved, thus affecting all groups in a similar manner.

## 1. Introduction

The COVID-19 pandemic, which started in December 2019 [1] and was declared as such by the WHO in January 2020 [2], was a major cause of health care system overload and resource occupation worldwide [3]. Chile was not exempt from this situation, where despite a successful vaccination program [4], hospitals consistently operated close to their maximum capacity during the first year of the pandemic [5].

Worldwide, different public health approaches were implemented to limit both the entry and the spread of the virus in the respective territories, being more or less successful, depending on the context, resources, and country preparedness [6]. Chile opted for a mixed strategy of active screening associated with general restrictive measures, implementing mass testing, together with dynamic quarantines, which were re-evaluated weekly [7,8].

The first case of COVID-19 in Chile was reported on 3 March 2020 [9], which gave the country an important window of time to prepare for the public health response. Then, in order to face the growing demand, all elective procedures were canceled, and new hospital facilities were promptly opened [10].

As the pandemic progressed and hospitals reached their maximum capacity, the facilities of various medical specialties began to be occupied to cover the still-growing demand from respiratory patients, leading to the cancelation of outpatient care for multiple specialties, including dermatology. Consistent with measurements taken in other countries, for the case of dermatology, residency activities were reorganized, chronic cases were predominantly followed up using teledermatology, and in-hospital interconsultations were limited to cases where diagnostic suggestions could not be achieved using teledermatology [11]. In addition, healthcare personnel was reassigned mostly to cover basic to intermediate wards. For outpatient consultations, similar to the experience in other countries [12], cases were first evaluated by a general practitioner and only urgent consultations were considered, using teledermatology as a tool to stratify the need for urgent specialist evaluation. The above-mentioned measures taken during the pandemic had important repercussions on waiting lists and access to specialized care for a population of patients who were displaced by the contingency [13]. Moreover, Chile is a country where there has been a historical limitation in access to dermatological care in the public system, with an important, documented gap between the current supply and demand of DC [14]. This is further deepened by the distribution of dermatologists across the country, with 64% of them concentrated in the capital city [15]. In addition, one-third of dermatologists work in the public sector, and thus there is stronger participation of dermatologists in the private sector [15]. Considering that the public system covers 80% of the population, and that dermatology accounts for 2.7% of the consultations to specialty care in the public health system [16], this distribution demonstrates wide inequities in the availability of dermatologists and access to dermatological care.

The reduction of dermatological consultation in the context of the pandemic and the restrictive measures taken in 2020 further deepen these gaps in access. This study is the first country-based analysis of the consultations made to specialty care during 2020 in Chile. It is important to establish this background and to evaluate which age groups and regions of the Chilean territory were most affected. This contributes to future development strategies to mitigate this acute limitation in access to care and establishes the groundwork for targeting the most affected population.

The aim of this study is to evaluate the impact of the COVID-19 pandemic on the total number of dermatology consultations (DCs) in the public health sector during 2020, evaluated according to geographic location, sex, and age range, and compared with consultations in the years before the pandemic.

## 2. Materials and Methods

The methods used were observational, cross-sectional, and retrospective studies of the consultations performed by dermatologists at the national level, in the Public Health sector, reported by the Department of Health Statistics and Information (DEIS) of the Subsecretaría de Salud Pública of the Chilean Ministry of Health (MINSAL) during 2020, compared with the available databases from 2017, 2018, and 2019 [17]. These consultations were contrasted with the demographic information available according to the 2017 Census and the population projections made by the National Institute of Statistics (INE) [18].

The incidence of DCs was calculated by age group and according to the population comprising those age ranges based on the 2017 Census data and INE projections. For the rest of the data, a univariate analysis was performed with measures of central tendency, percentage, mean, and standard deviation.

All analyses were performed using Stata software (Stata/SE 16.0 for macOS, Copyright 1985–2019 StataCorp LLC, College Station, TX, USA). All figures were developed using Microsoft Excel (Office 365, Microsoft Excel v16.66.1 for macOS, Copyright 1985–2022 Microsoft Crop, Redmond, DC, USA).

### Ethical Considerations

The data analyzed were obtained from a routinely collected anonymous database from the Department of Health Statistics and Information [17] and the National Institute of Statistics [18]. The analyses were conducted in compliance with the Declaration of Helsinki of ethical principles for medical research.

## 3. Results

A total of 830,724 DCs were recorded during the period 2017–2020. Of these, 120,095 were performed in 2020, with a reduction of 52.1% in 2020 as compared to 2019 (Figure 1).

Regarding the consultations according to sex, 59% (*n* = 71,387) corresponded to female patients, 41% (*n* = 48,700) corresponded to male patients, and 8 patients (0.0067% of the sample) had no information regarding sex.

Finally, the incidence of DCs in 2020 was 6.3 per 1000 population at the national level.

### 3.1. Consultations by Region during 2020

The Metropolitan Region, where the capital of Chile, Santiago, is located, accounted for 46% (*n* = 55,255) of the total number of DCs performed in 2020 (*n* = 120,095). Note that this region encompasses 42% of the total population of Chile [18]. When establishing the total number of consultations in terms of the population from each region of the territory, the region with the highest number of consultations per 1000 inhabitants is the Aysén region (Figure 2a). On the other hand, when establishing the incidence of consultations per 1000 inhabitants according to the geographical area (North, Center, or South), it is clear that the southern regions consult more than the northern and central regions of Chile (Figure 2b).

### 3.2. Consultations by Age Range in 2019 Versus 2020

During 2019, the age range with the highest number of DCs was 0–4 years (*n* = 18,300) and 15–19 (*n* = 18,197), while the lowest was 40–44 years (*n* = 11,142). In 2020, the age range with the highest number of DCs was 20–24 years (*n* = 8455), 15–19 years (*n* = 8274), and 25–29 years (*n* = 8256), accounting for 21% of the total number of DCs between 15 and 29 years. At the same time, the age range with the lowest number of DCs was 5–9 years (*n* = 5424) (Figure 3a).

The average reduction in DCs between 2019 and 2020 for each age range was 52 ± 5.1%, with a maximum reduction of 62.2% for age ranges 5–9 years and a minimum reduction of 42.5% for age ranges 30–34 years.

In terms of DCs by age range according to the population of the range, a maximum incidence was observed in the 75–79 age range, with 14.4 consultations per 1000 inhabitants. On the other hand, the age range with the lowest number of DCs as a function of its population was 40–44 years, with 4.2 consultations per 1000 inhabitants in the range (Figure 3b). 

Furthermore, when analyzing the proportions from the total consultations for a specific age group, it can be found that those proportions remain the same for the ranges from 50 to 54 (5.9% of the total sample), 55 to 59 (6.2% of the total sample), and 60 to 64 (6.3% of the total sample), with small variations of ±0.41% along the other age groups (Figure 3c).

Finally, when analyzing the incidence of DCs per 1000 inhabitants of a given age group for a specific year and dividing it by the average incidence of DCs per 1000 inhabitants of that year, the distribution shown in Figure 3d was observed.

### 3.3. Temporal Sequence of Consultations in 2020

When analyzing DCs month by month during 2020, a decreasing trend was observed after January, reaching a maximum minimum in April, one month after the first COVID-19 case was reported in Chile. From April onwards, a sustained growth trend was observed, even when, in June, the first peak of cases of the first wave of the pandemic occurred (13 June 2020) and when hospital capacity reached the peak occupation of ICU beds at the national level (2 July 2020) [19]. Finally, towards the end of the year, the growing trend of consultations tended to stabilize (Figure 4).

## 4. Discussion

The COVID-19 pandemic has had a notable impact on access to specialized health care worldwide. The closure of specialized ambulatory care, the need to allocate both economic and human resources, as well as the occupation of hospital units of different specialties to meet the growing demand for respiratory patients, had an important impact on diagnosis, treatment, procedures performed, and dermatology patient follow-up [20]. In 2020 in Chile, dermatology care in the public health system was reduced by more than half as compared to 2019 (52.1% reduction). This reduction can be correlated to the COVID-19 pandemic and the sanitary control measures, which restricted the movement of non-essential personnel and reallocated human resources and occupied available health care facilities to face the contingency, as previously mentioned. In contrast, it is reported that such factors are associated with the reduction in DCs by up to 80–90% [21].

Regarding the incidence of DCs, it can be seen that in 2020 it was 6.3 DCs per 1000 inhabitants. Comparing to 2019, with 14.1 DC per 1000 inhabitants [16], a reduction of more than half in the incidence of DCs between both years is observed. This is also consistent with the reduction observed in the total number of consultations in absolute terms. This documented reduction also has a notable impact on public health in the Chilean scenario. Note that, before the pandemic, waiting times for DCs in the public healthcare system were on average 341 days [22]. From this, it is expected that these waiting times have been extended even further in the context of the acute reduction in access to specialized care.

Concerning consultations by region, the Metropolitan Region accounts for the vast majority of DCs. However, when these consultations are contrasted according to the population of each region, it can be observed that the northern and southern regions had a higher incidence of DCs than the central regions (Figure 2). This may be because the gradual implementation of restrictive measures initially affected Santiago de Chile and the central regions of the country, which experienced the highest contagion rates at the beginning of the pandemic [7]. Furthermore, on 13 May 2020, a total quarantine was decreed for the Metropolitan Region. As cases increased in the central regions, dynamic quarantines were enforced based on the *Plan Paso a paso* implemented by the government at the time [19]. Nevertheless, when the cases started to increase in the northern and southern regions, the aforementioned restrictive measures were then extended to the rest of the country [19].

According to the results obtained, when analyzing the proportions of the consulting population by age range, these proportions remained relatively similar between 2019 and 2020, both in absolute numbers (Figure 3c) and concerning consultations per 1000 inhabitants (Figure 3d), with the difference that in 2020 the consulting population decreased significantly (Figure 3a). This is consistent with other studies, where, when comparing DC in 2019 to 2020, it has been reported that there were no significant differences concerning the mean ages of patients (46.26 ± 23.58 years vs. 47.06 ± 23.08), but a higher number of patients did not show up for DCs in 2020 [23].

As reported in the results section, the population aged 15 to 29 years led the total number of DCs, accounting for 21% of the sample (Figure 3a). However, when analyzing the incidence of DCs according to the population belonging to the age range, it was observed that patients aged 65 years and older led the dermatological consultations as a function of their population (Figure 3b). This is important to note, since at the time, attending a DC could risk exposure to COVID-19, and older adults are a known risk group for mortality due to COVID-19 [24]. In contrast, several studies have shown that worldwide, the average age of the consulting population was 40–50 years [25].

Regarding the distribution by sex, there was a predominance of consultations by the female sex. The results obtained in this area are consistent with the trend widely reported before the pandemic, where the approximate proportions were between 60–70% for females and 30–40% for males [16].

Regarding the temporal progression of the decrease in DCs (Figure 4), it can be observed that the month of April 2020 recorded the lowest number of DCs. This was possibly reactive to the confirmation of the first case of SARS-CoV-2 recorded in Chile on 3rd March of the same year [9] and the subsequent declaration of a state of catastrophe and constitutional exception, decreed by the government on 18 March 2020 [26]. Given the implications that such measures carry for both the general population and the health sector, this sharp drop in the total number of consultations was expected. In addition, this reduction is well documented in most studies evaluating the impact of the COVID-19 pandemic on DCs and could be due to the risk of SARS-CoV-2 transmission associated with outpatient care, and the postponement of elective consultations during the pandemic [27].

Nevertheless, as the pandemic progressed, dermatology consultations increased progressively and consistently. Even though June of 2020 recorded the highest number of cases up to that time, and during July of the same year ICU bed occupation reached the maximum nationwide capacity [28], such events did not seem to affect the growing trend of dermatology consultations in the following months. Moreover, this consistent increase could have been further fueled by the onset of new dermatological pathologies related to sanitary measures [29], and the stress caused by the confinement. Frequent handwashing and personal protective equipment usage generated an acute increase in contact dermatitis and other related dermatoses [30]. Furthermore, stress-related skin pathology could have started or worsened given the epidemiological scenario to which the population was exposed at the time. All of this could have impacted this dissociation between the epidemiological events and the increasing trend in consultations as the pandemic progressed.

Finally, it is worth highlighting the role that telemedicine could play in this context [31]. Teledermatology in Chile has been exponentially growing over the last decade. In December 2018, a unified teledermatology platform was implemented for the public health system, covering all primary care centers throughout the territory [32]. Chilean studies before the implementation of this platform have shown promising results in waiting times, training of general practitioners, improving resolution, and promoting timely access to care [33,34]. Thus, it can be implemented as an effective measure to provide dermatologic care to communities with limited access [35]. Along with this, international studies have shown encouraging results in delivering dermatological care [36,37], and its implementation in hospitals has been widely recommended due to the limitations of face-to-face care during the pandemic [38]. Therefore, teledermatology plays a central role in supplying the population’s demand remotely in the context of the pandemic [39], and also to optimize access and referral in the following years [40].

Finally, it is worth mentioning that this study has some limitations. First, it does not include diagnoses or the reason for consultation. From this, is not possible to determine whether the patients who consulted were following up previously diagnosed dermatologic pathologies or were first-time consultations. This has important implications in policy-making, as loss of follow-up or delay in diagnosis of potentially severe or malignant dermatologic pathologies could negatively affect the health of the consulting population. In addition, there is no information regarding the severity of the pathology or the resolution of the DC. Furthermore, there are no data regarding the consultations in the private sector at the time. The latter could not be determined, as the private sector is not required to have a centralized database of all the consultations performed by specialty care services.

Moreover, this study was based on results obtained during the initial period of the pandemic (2020), when restrictive measures prevailed as a public health strategy to mitigate COVID-19 infections. It does not include the later years (2021–2022) when there was a greater knowledge of the virus and a considerable proportion of the population was vaccinated. In view of this, it would be prudent to propose new studies on the delay in diagnosis and treatment of chronic or malignant dermatologic pathologies secondary to the acute deficit in access to dermatologic care that occurred during 2020, as well as to compare these data with the distribution of DCs during 2021 and 2022 and consider the impacts that it might have.

## 5. Conclusions

The COVID-19 pandemic has had a great impact on the total number of dermatology consultations in the Chilean public health sector, drastically modifying the discrete increasing trend in the total number of consultations observed from 2017 to 2019. Thus, during 2020, these consultations were drastically reduced to more than half, consistent with reports from other studies. Finally, despite the restrictions and difficulties in access during the pandemic months in 2020 in Chile, the consulting population that accessed dermatologic care did so in similar proportions in terms of sex and age range. 

## Figures and Tables

**Figure 1 healthcare-11-00633-f001:**
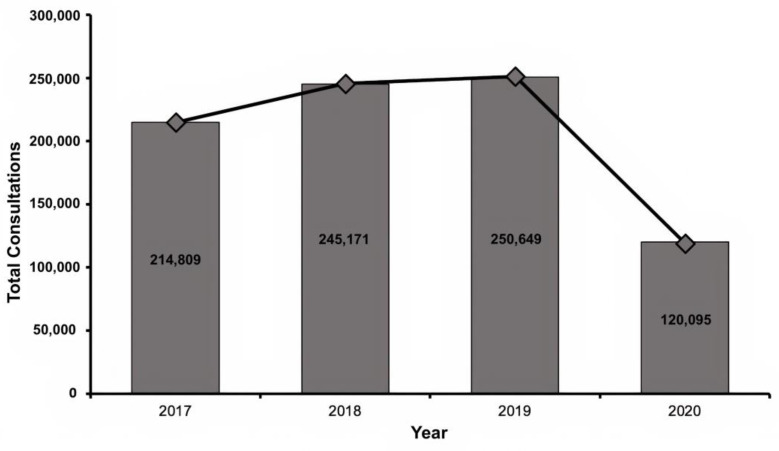
Total dermatology consultations in the public health sector from 2017 to 2020.

**Figure 2 healthcare-11-00633-f002:**
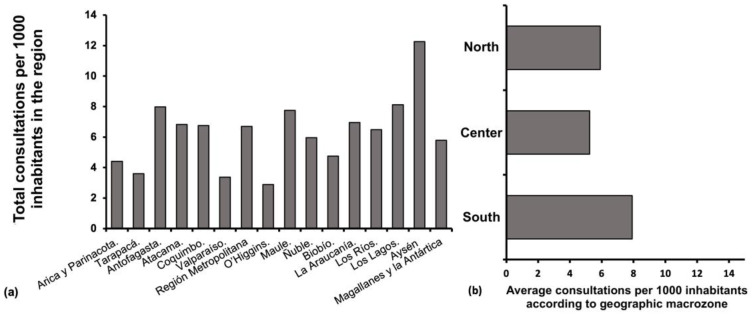
(**a**) Dermatology consultations in Chile’s public sector per 1000 inhabitants in each region, ordered from north to south from left to right. (**b**) Consultations to dermatology according to the average number of consultations per 1000 inhabitants by geographic zone (North, Center, and South) North zone covers the regions of Arica and Parinacota to Coquimbo. The central zone covers the regions of Valparaíso to Bío Bío. The South zone covers La Araucanía to Magallanes and Antarctica regions.

**Figure 3 healthcare-11-00633-f003:**
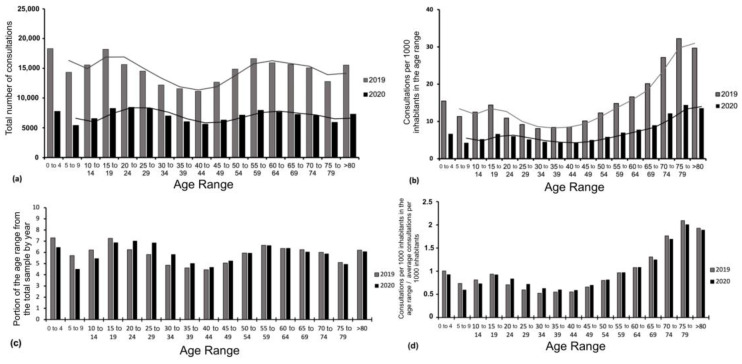
Dermatology consultations by age range according to total consultations and (**a**) according to the number of consultations per 1000 inhabitants in the age range; (**b**) their respective average trend lines, for the years 2019 and 2020; (**c**) dermatology consultations by age range with respect to the total consultations for the year; and (**d**) the ratio between the incidence of dermatology consultations per 1000 inhabitants of a specific age range and the average incidence of dermatology consultations per 1000 inhabitants.

**Figure 4 healthcare-11-00633-f004:**
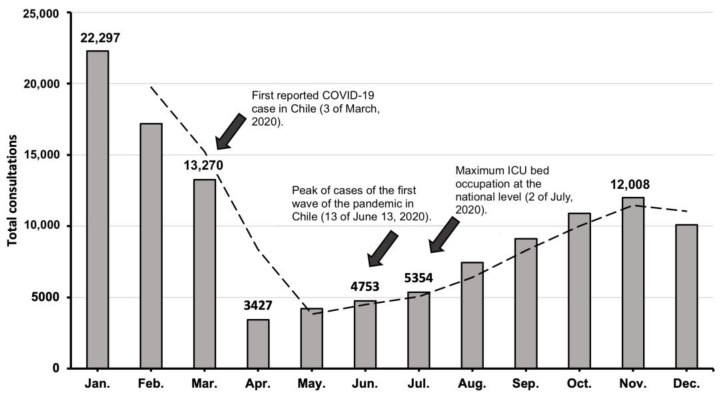
Monthly temporal progression of dermatology consultations during 2020 and its correlation with the epidemiological situation of the country at the time [9,19].

## Data Availability

All data from this study is in the public domain and can be accessed at the DEIS and INE websites of the Chilean government.

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
