# Peer review of "Impact of the COVID-19 Pandemic on Dermatology Care in the Chilean Public Health Sector"

_healthcare, 2023, doi:10.3390/healthcare11050633_

Round 1

Reviewer 1 Report

Dear Authors,

I would like to say that your paper is interesting and really well done. 
In my point of view, in your study there is a lack a limitations which currently are important part of every scientific paper.

I would like to also ask you for the statement in the paper if there are similar studies done in dermatology field from the another countries or in the another medicine specialities. It would be interesting for the readers to know how the statistics related to pneumonology or cardiology looked like that time. If there is no such studies in the another fields, please add the statement „To the best of our knowledge there is no other similar papers from the another specialities”

169 Line - it would be better to write „Comparing to 2019” 

Author Response

Dear Reviewer,

In the attachment, you will find the response to the comments. Thank you very much.

Reviewer 2 Report

Dear Authors, you paper is rich of information about epidermiology and DC during 2020, the year of the great covid pandemic. Nevertheless, the great limitation is the loss of diagnoses.

Italy was the 1st European country where a covid parient was identified. Healthcare personnel was completely dedicate to fight pandemic in other wards (Nazzaro G, et al.. What is the role of a dermatologist in the battle against COVID-19? The experience from a hospital on the frontline in Milan. Int J Dermatol. 2020 Jul;59(7):e238-e239). In Chile what happened ?

Only urgent DC was authorized such as in other countries ? (See Giacalone S, et al. Which are the "emergent" dermatologic practices during COVID-19 pandemic? Report from the lockdown in Milan, Italy. Int J Dermatol. 2020 Aug;59(8):e269-e270)

Author Response

(The authors gave the same response as above.)

Reviewer 3 Report

The authors have done a credible job of assessing the COVID pandemic on public healthcare services provided for dermatology consultations in their country. These same types of results have been demonstrated in other sectors of medical care in the world.

The authors should clarify in their country what proportion of the dermatology care is provided by the public as compared with the private sector. They do make it clear that they could not track the same outcomes in the private sector.

Author Response

(The authors gave the same response as above.)

Reviewer 4 Report

Manuscript title: Impact of the COVID-19 pandemic on dermatology care in the Chilean Public Health Sector 

Authors: Aragón-Caqueo et. al

Recommendation

Minor Revision

Wold you be willing to review a revision of this manuscript?

No

Comments to the Authors:

(A) Provide an overview/summary of the manuscript

​This paper is a study that evaluates the impact of the pandemic on dermatology care in the Chilean Public Health Sector. The total number of dermatological consultations (DC) to the Chilean public sector in 2020 is analysed according to sex, age range and compared with the available databases from 2017 to 2019. 

​The topic of this article is of maxim interest since due to the Coronavirus-19 (COVID-19) pandemic, most resources of the public health systems were allocated to the increasing demand from respiratory patientsFrom this, it is expected that specialty consultations would decrease drastically. 

​Thus, this article is very useful and of biggest interest for the specialists who take care of such patients.

​The manuscript is structured in five parts:

1. Introduction 2. Materials and Methods 3. Results 4. Discussion 5. Conclusions

(B) Introduction

​Aragón-Caqueo et. al hilighted the huge impact of the COVID-19 pandemic on the other specialties, including Dermatology, especially in Chile, where there already was a wide inequity in the availability of dermatologists and access to dermatological care.

​The aim of this study was again, hilighted, to evaluate the impact of the COVID-19 pandemic on thetotal number of dermatology consultations (DC) in the public health sector during 2020, evaluated according to geographic location, sex, and age range, and compared with consultations in the years before the pandemic. 

​The introduction is well documented and sets the context for the further information. It ends with a statement of purpose: It is important to establish this background and to evaluate which age groups andregions of the Chilean territory were most affected. This contributes to the future development strategies to mitigate this acute limitation to access to care and establishes the groundwork for targeting the most affected population..

(C) Materials and Methods

​Materials and Methods are accurately described. The analysis was conducted in compliance with the Declaration of Helsinki of ethical principles for medical research. 

(D) Results

​The results are concise and are supplemented with four key charts. 

​The quality of the data presented is good and the results appear to be valid.  This chapter is divided in multiple sections: consultations by sex, consultations by region during 2020, consultations by age range in 2019 versus 2020 and temporal sequence of consultations in 2020.

​The results reflect the methods in organisation and structure.

(E) Conclusions and Discussion

​The Discussion and Conclusions support the data being presented.

​The last paragraph of the Discussion section includes the known limitations of the data and the analysis.

​In the Conclusion section, the authors clearly stated what they have identified in their research (“…during 2020, these consultations are drastically reduced to more than half, consistent with reports from other studies. Finally, despite the restrictions and difficulties in access during the pandemic months in 2020 in Chile, the consulting population that accessed dermatologic care did so in similar proportions in terms of sex and age range.).

(F) Quality of English language

​Overall, the manuscript should be accepted, after a minor revision of English language and style.

Author Response

(The authors gave the same response as above.)
